# Experimental evidence for core-Merge in the vocal communication system of a wild passerine

Toshitaka N. Suzuki [1,2] & Yui K. Matsumoto[2,3]

One of the cognitive capacities underlying language is core-Merge, which allows senders to combine two words into a sequence and receivers to recognize it as a single unit. Recent field studies suggest intriguing parallels in non-human animals, e.g., Japanese tits (*Parus minor*) combine two meaning-bearing calls into a sequence when prompting antipredator displays in other individuals. However, whether such examples represent core-Merge remains unclear; receivers may perceive a two-call sequence as two individual calls that are arbitrarily produced in close time proximity, not as a single unit. If an animal species has evolved core-Merge, its receivers should treat a two-call sequence produced by a single individual differently from the same two calls produced by two individuals with the same timing. Here, we show that Japanese tit receivers exhibit antipredator displays when perceiving two-call sequences broadcast from a single source, but not from two sources, providing evidence for core-Merge in animals.

It has been hypothesized that the generative power of language is derived from a cognitive capacity called "Merge"[1–3]. No matter how the meaning is created, Merge allows senders to combine two linguistic items (e.g., two words or two phrases) into larger sequences and receivers to recognize it as a single unit[1–3]. The most basic form of Merge is often specifically referred to as "core-Merge" in which two words are combined to form a new unit (e.g., come + talk = {come talk}, a + dog = {a dog})[2,3]. Although once considered a uniquely human capacity, recent field studies suggest intriguing parallels with core-Merge in non-human animals (hereafter animals): several species of birds and mammals combine two call types, each with their own meaning, into larger sequences which evoke specific behavioural responses in receivers that are different than their responses to each component call type[4–6]. However, there is an alternative explanation for these animal examples that does not depend on core-Merge: receivers may perceive a two-call sequence as two individual calls that are arbitrarily produced in close time proximity (i.e., temporally linked), not as a single unit[5,7–9]. Due to the lack of studies examining whether animals perceive a two-call sequence as a single unit or not, it

remains unknown whether core-Merge is unique to humans or whether it has also evolved in non-human species.

In this study, we develop a novel paradigm to test whether animals use core-Merge to interpret two-call sequences (i.e., if they recognize two combined call types coming from a single individual as forming a single unit) or whether they respond to any temporally linked calls (i.e., arbitrarily produced or not, coming from one individual or more) in the same way. To do this we propose the following single-sender/multiple-sender paradigm. If an animal has evolved core-Merge, then it should be able to distinguish a two-call sequence produced by a single individual (i.e., combined calls) from two temporally linked calls produced by multiple individuals (i.e., non-combined calls). In other words, animal receivers should recognize whether two temporally linked calls are produced from the same spatial location (see Fig. 1 for a human example). On the other hand, if core-Merge does not operate, then the temporal linkage of two calls should be sufficient to evoke specific responses; receivers should show similar responses to two calls being produced by one or two sources, as long as they are temporally linked. Using one- and two-speaker playbacks, we can test

[1]The Hakubi Center for Advanced Research, Kyoto University, Yoshida-honmachi, Sakyo-ku, Kyoto 606-8501, Japan. [2]Department of Zoology, Graduate School of Science, Kyoto University, Kitashirakawa-oiwake-cho, Sakyo-ku, Kyoto 606-8502, Japan. [3]Department of Information Medicine, National Institute of Neuroscience, National Center of Neurology and Psychiatry, 4-1-1, Ogawahigashi-cho, Kodaira, Tokyo 187-8502, Japan. e-mail: toshi.n.suzuki@gmail.com

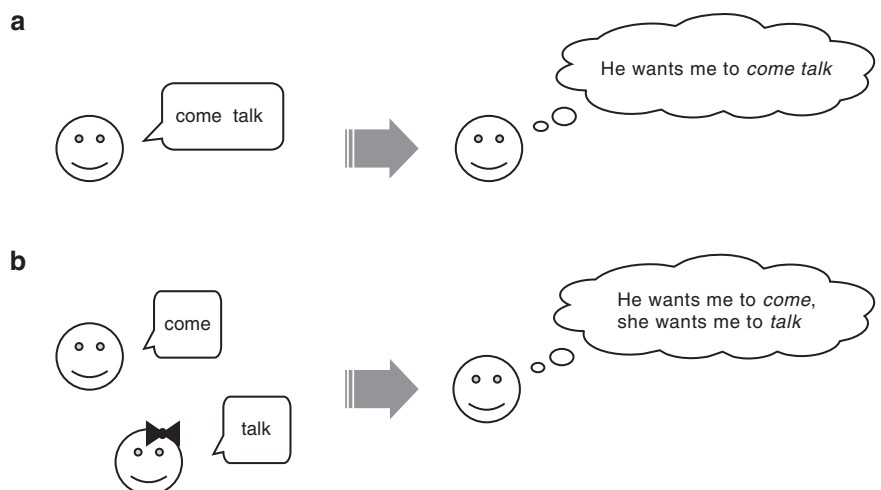

**Fig. 1 | Experimental paradigm for testing core-Merge. a** In language, core-Merge allows receivers to recognize two temporally liked words as a single unit (e.g., "come" + "talk" = {come talk}). **b** However, if the same words are separately given by two persons, receivers may perceive them as two individual messages (e.g., "come" from one person and "talk" from the other). If an animal species has evolved core-Merge, then receivers' responses to two temporally linked calls should depend on whether the two calls are produced by a single individual.

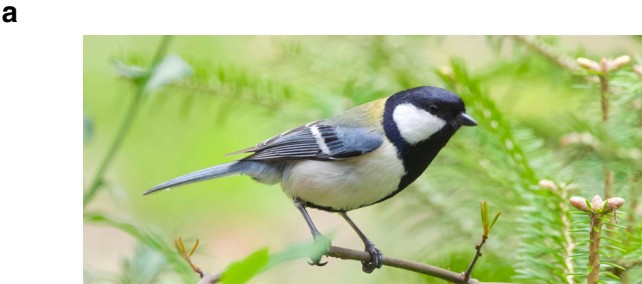

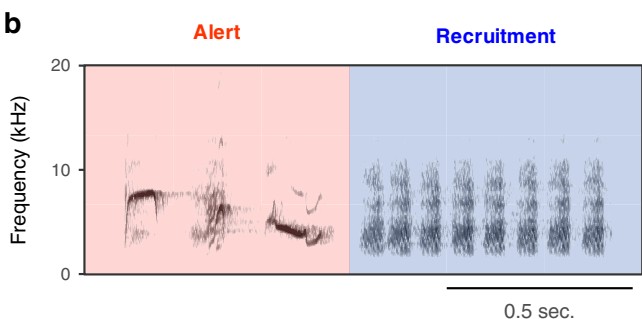

**Fig. 2 | Study system.** Japanese tits (**a**) combine alert and recruitment calls into alert-recruitment call sequences (**b**) when gathering other individuals to mobbing of a predator.

whether receivers' responses to two temporally linked calls depend on whether they are produced by a single sound source, and, if so, this provides evidence for core-Merge.

We use this paradigm to explore core-Merge in a wild bird species, the Japanese tit (*Parus minor*) (Fig. 2a). These birds produce alert calls when warning conspecifics about danger, such as the presence of predators, while they produce acoustically distinct recruitment calls when attracting conspecifics to non-dangerous situations, such as food locations or during nest visitations[10,11]. They often combine these call types into ordered sequences (alert-recruitment call sequences) when gathering other individuals to approach and harass (i.e., mob) a predator[10] (Fig. 2b). Previous experiments showed that tits display

different behaviours when hearing alert calls (moving their head horizontally as if scanning for danger) and recruitment calls (approaching the sound source)[11]. In response to alert-recruitment call sequences, tits progressively approach the sound source while continuously scanning the horizon, suggesting that they detect compound information (i.e., "alertly" + "approach")[11]. However, if the call order is artificially reversed, tits reduce their response, indicating that they perceive whether the component calls are temporally linked into specific sequences[11].

Based on these findings, we hypothesized that Japanese tits have evolved core-Merge and recognize an alert-recruitment call sequence as a single unit. If this is the case, then tits are expected to exhibit appropriate responses to alert-recruitment call sequences given by a single individual; however, they should not perceive the same information when alert calls and recruitment calls are separately produced by two individuals. To test this prediction, we exposed free-living flocks of Japanese tits to (i) alert-recruitment call sequences broadcast from a single speaker and (ii) alert calls and recruitment calls broadcast from two speakers in turn, following the alert-recruitment order (Fig. 3a, b). To ensure that the differences between the two treatments only depend on the number of speakers, we created all the sound files using the same procedure; we copied alert calls and recruitment calls separately from Japanese tits' natural call sequences, and then pasted them onto background noise files, making the intervals between these call types constant (0.1 s) across treatments. We created 90-s-long playback stimuli containing the same number of alert and recruitment calls (30 calls for each call type) at a natural calling rate (one call per 3 s for each speaker).

Upon finding a flock, we placed one or two Bluetooth speakers (SoundLink Micro, Bose) on tree branches. In treatments with two speakers, we separated them by 10 m, which is a natural distance between two individuals within a flock. We also placed a taxidermic specimen of bull-headed shrike (*Lanius bucephalus*) on a tree branch 5 m from the speaker(s) in a natural perching posture (Fig. 3). Bull-headed shrikes are a major predator of small passerines, and tits often approach and harass them with wing flicking displays (i.e., mobbing)[12,13]. Exposure of a predator specimen in combination with playback stimuli allowed us to measure tits' mobbing behaviours during one- and two-speaker playbacks through a common standard. During 90-s of playback, we recorded (i) the percentage of individuals in Japanese tit flocks that approached within 2-m of the shrike

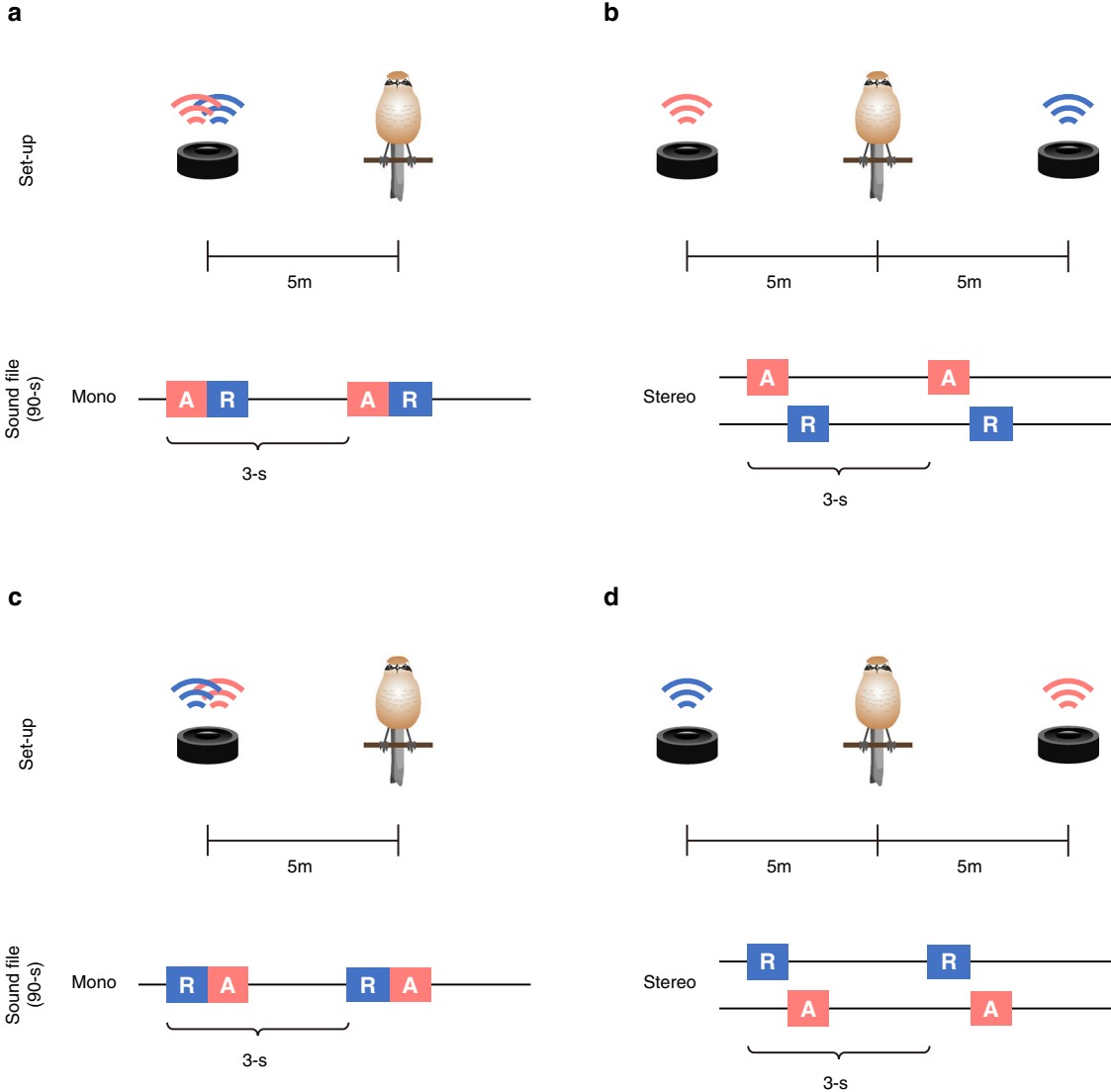

**Fig. 3 | Experimental set-up and sound files.** If an animal uses core-Merge to perceive call sequences, then it should be able to assess whether the component calls are produced from the same spatial location, as well as whether they are temporally linked into naturally ordered sequences. Japanese tits are exposed to a shrike specimen in combination with four types of playback stimuli: **a** alert calls and recruitment calls are broadcast from one speaker as temporally linked, alert-recruitment sequences, **b** the same two calls are broadcast from two speakers, while they are temporally linked, **c** recruitment calls and alert calls are broadcast from one speaker, but they are not naturally ordered, **d** the two calls are not linked in either space or time. In two-speaker treatments, the speakers and a shrike specimen were placed in a straight line.

specimen and (ii) the percentage of flock members that exhibited wing flicking displays.

Here, we show that Japanese tits mob a shrike specimen when hearing alert-recruitment call sequences played from a single speaker, but not when hearing the same two calls played from different speakers with the same timing. This demonstrates that tits recognize an alert-recruitment call sequence produced by a single individual as a single unit, and not merely as two temporally linked calls, providing evidence for core-Merge in a non-human species.

## Results

### Do tits recognize a call sequence as a single unit?
Japanese tits responded differently to the shrike specimen during one- and two-speaker playbacks (Fig. 4). During the one-speaker playback of alert-recruitment call sequences, tits typically approached within 2 m of the shrike and exhibited wing flicking displays (Supplementary Movie 1). However, when alert and recruitment calls were separately broadcast from two speakers, tits rarely mobbed the shrike: they

infrequently approached it and rarely exhibited wing flicking displays (generalized linear mixed model: approach: $Z = 5.50$, $P < 0.0001$; wing flicking: $Z = 5.68$, $P < 0.0001$). Therefore, tits' responses do not merely depend on the alert and recruitment calls being temporally linked, but rather on their perception of the sequence being broadcast from a single source. This supports the hypothesis that tits perceive an alert-recruitment call sequence as a single unit produced by a single individual.

### Does tits' mobbing depend on temporal linkage of two calls?
Although a previous study showed that temporal linkage of two calls (call ordering) influences tits' behavioural responses[11], there remains the possibility that, in the presence of a shrike specimen, simply hearing two call types from a single source causes tits to exhibit mobbing behaviour. If this is the case, then tits are expected to mob the shrike when hearing any sequences of alert and recruitment calls, as long as they are produced by a single source. To account for this possibility, we exposed flocks to artificially reversed, recruitment-alert

**a**

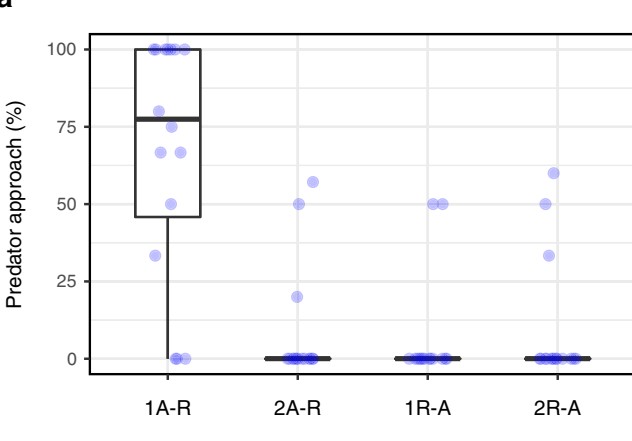

**b**

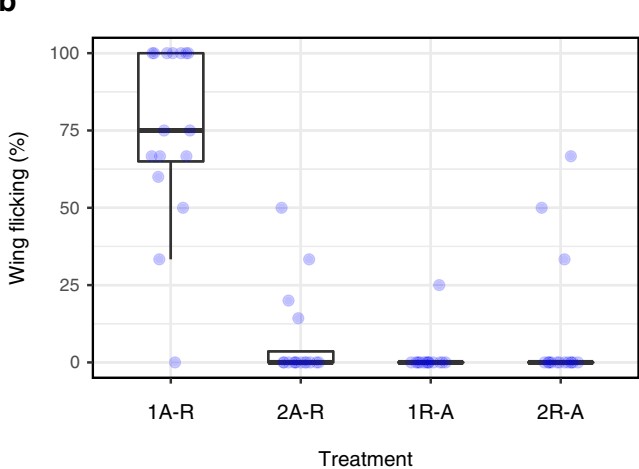

**Fig. 4 | Predator mobbing by Japanese tits during call playbacks. a** Percentage of individuals in Japanese tit flocks that approached within 2-m of the shrike specimen (generalized linear mixed model: $\chi^2 = 80.16$, $df = 3$, $P < 0.0001$). **b** Percentage of individuals in Japanese tit flocks that exhibited wing flicking displays ($\chi^2 = 95.75$, $df = 3$, $P < 0.0001$). The box-and-whisker plots display the median, 1st and 3rd quartiles; the whiskers are extended to the most extreme value inside the 1.5-fold interquartile range. Statistical significance was calculated using two-sided log-likelihood ratio tests. Sample size: $n = 16$ flocks for each treatment, resulting in $n = 64$ flocks across all four treatments. 1A-R one-speaker playback of alert-recruitment sequences, 2A-R two-speaker playback of alert calls and recruitment calls arranged in this order, 1R-A one-speaker playback of recruitment-alert sequences, 2R-A two-speaker playback of recruitment calls and alert calls arranged in this order. See Supplementary Table 1, Supplementary Table 2, and Supplementary Fig. 1 for details of statistical analyses. Source data are provided as a Source Data file.

call sequences broadcast from one speaker (Fig. 3c). The calling rate and the duration between two call types were identical to those of one-speaker playback of alert-recruitment sequences, but the call types were not presented in the naturally ordered sequences. Tits exhibited weaker responses to a shrike during one-speaker playback of recruitment-alert sequences than during one-speaker playback of alert-recruitment sequences (approach: $Z = 5.10$, $P < 0.0001$; wing flicking: $Z = 4.69$, $P < 0.0001$; Fig. 4), indicating that they are sensitive to temporal linkage of two calls even in the presence of a predator.

We further exposed flocks to recruitment calls and alert calls separately broadcast from two speakers in this order (Fig. 3d), so that the component calls are not linked in either time or space. Tits rarely mobbed the shrike specimen during two-speaker playback of

recruitment-alert call ordering (Fig. 4), which was significantly different from responses during one-speaker playback of alert-recruitment call sequences (approach: $Z = 4.90$, $P < 0.0001$; wing flick: $Z = 5.54$, $P < 0.0001$). Further pairwise comparisons reveal that tits exhibit predator mobbing when and only when they perceive naturally ordered, alert-recruitment call sequences produced by a single source (see Supplementary Table 1). These results show that tits' mobbing responses depend on both whether the two calls are temporally linked and whether they are produced from the same spatial location.

### Do any other factors influence tits' behaviour?

We carefully designed the experiments to control for the possibility that factors other than temporal and spatial linkages of the two call types may influence tits' mobbing responses. First, we controlled for the possibility that subtle variation within each call type may provide information about callers' identity, which might influence receivers' responses. We prepared 16 unique sets of alert and recruitment calls using either calls from the same bird ($n = 8$ source individuals, $n = 8$ call sets) or from two different birds ($n = 16$ source individuals, $n = 8$ call sets). Then, we created 64 playbacks from the 16 call sets in which each call set was used to construct four playback treatments for a block ($n = 16$ blocks; e.g., block no. 2: alert call from bird no. 2 and recruitment call from bird no. 17 were played together from the same speaker, from different speakers, and in reversed order from the same speaker and from different speakers; Supplementary Table 3). As expected, there was no significant influence of the number of source individuals (one or two) on tits' mobbing responses (approach: $\chi^2 = 0.78$, $df = 1$, $P = 0.3777$; wing flicking: $\chi^2 = 0.69$, $df = 1$, $P = 0.4046$).

Second, we also controlled for the possibility that social context may influence the willingness of tits to join in mobbing. Since flock size has been suggested to increase the intensity of mobbing[14,15], we recorded the number of Japanese tits observed around 15-m of the shrike specimen during 90-s of playback and included this as a covariate in the statistical models. Supporting the prediction, tits more readily approached within 2-m of the shrike when flock size was larger (generalized linear mixed model: $\chi^2 = 16.06$, $df = 1$, $P < 0.0001$). However, they did not alter wing flicking displays according to the flock size ($\chi^2 = 0.00$, $df = 1$, $P = 0.9692$). Approaching a predator is likely to be riskier than exhibiting wing flicking displays, but the associated risks should be reduced when there are more surrounding individuals (i.e., safety in numbers)[16]. This might explain why flock size affected approaching behaviour, but not wing-flicking.

## Discussion

Our results show that Japanese tits discriminate between two temporally- and spatially linked calls played from one speaker (which mimic calls by one individual) and two temporally linked calls played from two speakers (which mimic calls from two individuals). They join in mobbing a shrike when perceiving alert-recruitment call sequences broadcast from a single sound source (i.e., combined calls). In contrast, if the component calls are broadcast separately from different sources in the same ordering (i.e., non-combined calls), tits reduced their mobbing response. During playbacks of recruitment-alert orderings from one and two sources, tits rarely mobbed the shrike, indicating that they recognize whether the two calls are temporally linked into ordered sequences even in the presence of a predator. These results are supported by the statistical models that control for the possibilities that other factors, such as the way of creating playback stimuli and flock size, may have influence on tits' behaviour. These findings show that tits are able to recognize an alert-recruitment call sequence as a single unit when coming from one individual, but not from two, which supports our conclusion that tits have evolved core-Merge.

Previous experiments showed that Japanese tits do not simply associate an alert-recruitment call sequence with an independent meaning, such as "mobbing", but rather, extract meanings of both

component calls (i.e., "alertly" + "approach" = {alertly approach})[11,17]. Therefore, call combinations of Japanese tits might represent an analogy to human phrases where core-Merge operates on two words to produce juxtaposed, compositional phrases (e.g., come + talk = {come talk}). Combinations of two call types have also been documented for other animals; however, how they relate to the creation of meaning seems to be diverse across species[4–6]. For example, putty-nosed monkeys (Cercopithecus nictitans) combine two alarm call types, each of which seemingly denotes a different predatory threat, such as leopards or eagles, to stimulate long-distance group movements[18,19]. Since this combination creates a message that is not derived from either alarm call type, it might represent an analogy to idiomatic expressions, i.e., "leopard" + "eagle" = "move on"[20] (but see ref. 21). Campbell's monkeys (Cercopithecus campbelli) add a short vocal element at the end of high-threat alarm calls when perceiving lower threats, which has thought to be an analogy to suffixation (i.e., "predator" + "-like" = "predator-like")[22,23]. Regardless of how meaning is created, the production and perception of animal call combinations may largely depend on core-Merge. We hope that our experimental paradigm provides a robust method to investigate core-Merge across a variety of species and encourages future comparative studies, which will help to understand under which conditions this linguistic capacity will evolve.

This study not only provides evidence for core-Merge in animal communication systems, but also has important implications for the studies of language evolution. There are two conflicting theories for the origins of language's productivity. One theory holds that a single cognitive capacity called "Merge" enables us to produce and comprehend any kind of word combinations, including complex expressions with hierarchical structure (e.g., a + dog + barks = {a dog} + barks = {{a dog} barks})[24,25]. The second theory holds that such complex expressions require, in addition to Merge, another cognitive capacity called "recursion" that allows us to form hierarchically structured mental representations[2–5,26–28]. In this theory, it is expected that without recursion, Merge merely serves to combine two words, which is often labelled as core-Merge[2]. In other words, the combination of core-Merge and recursion enables fusing more than two words into hierarchical expressions, which is referred to as recursive-Merge[2]. Our findings support this second theory, since tits combine two call types into a single unit, but show no evidence that they produce sequences with more than two meaningful calls; further research is necessary to determine if tits can create hierarchically structured sequences. We stand at a starting point to explore the similarity and difference of the combinatorial communication systems between animals and humans[3–9,29]. Determining how widely Merge is involved in animal signals and what specific mechanisms provide the basis for the emergence of hierarchical structure remains a key challenge in animal communication research, which will deepen our understanding on the evolutionary pathway of language.

## Methods

### Study site and animals

We studied $n = 64$ flocks of Japanese tits in mixed deciduous-coniferous forests in Nagano and Gumma (36°17-31'N, 138°26-39'E), Japan. Although most of the birds had not been individually colour-ringed, all the experimental trials were conducted at least 400 m apart; previous observations on colour-ringed individuals showed that this distance was enough to ensure the collection of data from different individuals[30]. In this site, one of the major predators of small birds is the bull-headed shrike, which is often mobbed by small birds including Japanese tits.

### Playback stimulus

To test whether Japanese tits recognize an alert-recruitment call sequence as a single unit, we prepared four treatments: (i) one-speaker playback of alert-recruitment call sequences, (ii) two-speaker playback

of alert-recruitment call sequences with alert and recruitment calls played from different speakers, (iii) one-speaker playback of recruitment-alert call sequences, (iv) two-speaker playback of recruitment-alert call sequences with recruitment and alert calls played from different speakers (Fig. 3). We created sound files for these treatments using the software program Audacity 2.1.3 (http://www.audacityteam.org). For one-speaker treatments, we composed mono sound files where call sequences were repeated onto a single channel, whereas for two-speaker treatments, we composed stereo sound files where either alert or recruitment calls were repeated onto the right or left channels, respectively. All the files contained an equal number of alert calls (30 calls) and recruitment calls (30 calls) at the same rate (one call every 3 s), resulting in 90-s of stimuli (Fig. 3), which corresponds to the range of the natural calling rate of alert-recruitment sequences during mobbing by Japanese tits[10]. For all stimuli, within-call-sequence intervals between alert and recruitment calls were constant (0.1 s), which is within the range of intervals of these calls in natural call sequences[11,17]. In contrast, between-call-sequence intervals varied from 1.50 to 1.81 (median = 1.68) due to the difference in call length, but were constant across playback stimuli within the same "block" where the four treatments were created using the same call exemplars (see below). While alert calls are composed of three distinct note types, recruitment calls are strings of the same note type that vary in repetition number. Since the repetition number can vary depending on predator type[10], we conducted predator exposure experiments to Japanese tit flocks ($n = 12$) and recorded call sequences towards a bull-headed shrike life-like specimen. In response to a shrike specimen, tits produced alert-recruitment call sequences with a recruitment note repetition number ranging from 5 to 15. Since the interquartile range of repetition number was 6.75 to 10, we used recruitment calls with 7–10 notes as playback stimuli in this study. In consideration for the possible influence of sound editing procedure, we created all the stimuli in the same manner; we copied alert and recruitment call parts separately from recording files, and pasted them onto background noise files to produce all four types of stimuli. Playback amplitudes were constant across treatments, 70 dB at 1.0 m measured using a sound level meter (SM-325, AS ONE Corporation). Therefore, the differences between treatments only depend on whether these calls are produced as sequences from the same source and how the calls are ordered.

We carefully designed experiments to control for the possibility that individual-based acoustic features in alert and recruitment calls might influence tits' responses. First, we prepared 16 unique sets of alert and recruitment calls using either calls from the same bird ($n = 8$ source individuals, $n = 8$ unique call sets) or from two different birds ($n = 16$ source individuals, $n = 8$ unique call sets). Then, we created the four types of treatments (i.e., alert-recruitment call sequences from the same speaker, from different speakers, and in reversed order from the same speaker and from different speakers) from each of the alert-recruitment call sets, resulting in 16 blocks of playback stimuli (Supplementary Table 3). This allows us to test the possible influence of individual-based acoustic variation on receivers' responses.

We were also careful to avoid the possible influence of population-level signatures of acoustic features: we only used Japanese tits' call sequences that had been previously recorded from the same study population. We saved the sound files in .wav format (16-bit accuracy, 48-kHz sampling rate) onto a playback device (iPhone 8, Apple Inc.). We used the default Music app (Apple Inc.) to playback the sound files.

### Experiment

We (TNS and YKM) conducted experimental trials from 26 October to 4 December 2020 and during the period of 0800 and 1600 h (Japan Standard Time). We did not conduct trials under wet and windy weather conditions, since these may influence behavioural patterns of forest birds[31]. First, we searched for and located a flock of Japanese tits.

Upon finding a flock, we fixed a taxidermic specimen of bull-headed shrike in a perching posture on the branch at 1.8 ± 0.2 m (mean ± s.d., $n = 64$) above the ground. Then, we placed either one or two Bluetooth speakers (SoundLink Micro, BOSE) on tree branches at 1.6 ± 0.2 m (mean ± s.d., $n = 96$) above the ground, and oriented them upwards to control for the possible influence of directionality. We set the distance between the shrike specimen and the speaker(s) at 5 m. For trials with two speakers, we set the distance between speakers at 10 m, mimicking the situation in which two birds are calling (Fig. 3). The shrike specimen was first covered with a black cloth and was exposed by removing the cloth just before each trial.

We began playbacks when at least two Japanese tits were present within 15 m from the shrike specimen. During 90-s of playbacks, we recorded (i) whether birds approached within 2-m of the shrike specimen during the playback and (ii) whether birds exhibited wing flicking displays[12,13]. We counted the number of individuals within 15 m from the shrike during 90-s of playbacks and considered it as flock size. During trials, we sat on the ground at ca. 10 m from the shrike specimen to decrease the influence of the observers' presence on bird behaviour. To account for the inter-observer reliability[32], we calculated intra-class correlation coefficient (ICC; *icc* function in the R package *irr*) between us. The lowest ICC was 0.998, indicating high degree of inter-observer reliability for the two behavioural measurements. We also video-recorded the responses of tits using a digital video camera (FDR-AX60, SONY). After completion of each trial, we checked the video recording and made an on-the-spot confirmation of the exact location at which each bird made the closest approach to the shrike specimen during the 90-s of playbacks. Then, using a tape measure, we recorded the minimum approach distance of birds to the shrike specimen. Thus, our final data set consisted of the most reliable observations confirmed by two experimenters and video evidence.

The order of trials was randomized within each block ($n = 16$ blocks), each of which is composed of a unique alert-recruitment call set but includes four treatments differing in the number of speakers and call order. Therefore, responses to all four treatments were observed under largely similar conditions. In a few trials, the first bird to approach the shrike specimen was from a heterospecific species, such as a varied tit ($n = 1$) or a long-tailed tit ($n = 1$). To account for the possibility that these birds evoke mobbing behaviour in Japanese tits, we only used the data from instances where the first individual to approach the shrike was a Japanese tit. Otherwise, we repeated the same treatment at a different site.

We used 64 unique playbacks created from 16 unique sets of alert-recruitment calls for 64 trials in order to avoid pseudoreplication[33]. We prepared two specimens of male bull-headed shrikes and used each of them for the equal number of trials. We did not use specimens of female shrikes since females migrate from the study site in late summer and only males were observed during the study period.

### Statistical analysis

We analyzed the effect of playback treatments on the mobbing behaviours of Japanese tits using generalized linear mixed models in R[34,35]. We used the proportions of Japanese tits in flocks that (i) approached within 2-m of the shrike specimen and (ii) exhibited wing flicking displays. For the analysis of predator approach, we prepared two vectors (i.e., the number of Japanese tits that approached the shrike specimen and the number of Japanese tits that did not approach the shrike specimen). Then, we created a single response variable by binding together these two vectors using *cbind* function. Similarly, for the analysis of wing flicking displays, we created a single response variable by binding two vectors (i.e., the number of tits that exhibit wing flicking and the number of tits that did not exhibit wing flicking). We fitted playback treatments as a fixed term, and flock size (maximum number of Japanese tits observed during 90-s of playback) and the way

of creating playback stimuli (whether the two call types were recorded from a single individual or two individuals) as covariates. We also included identity of alert-recruitment call sets that were used for creating playback stimuli (i.e., call sets from either one or two source individuals) and identity of shrike specimens as random terms. We used a binomial error distribution and logit-link function (*glmer* in the R package *lme4*) for these models. Statistical significance was calculated by log-likelihood ratio tests using *anova* in the R package *stats*. We further conducted post-hoc pairwise comparisons between treatments by using estimated marginal means (*emmeans* in the R package *emmeans*). When making pairwise comparisons, we adjusted *p*-values by applying a false discovery rate control for multiple testing[36]. All tests were two-sided and the significance level was set at $\alpha = 0.05$. Exact *p*-values are reported when $p \geq 0.0001$.

### Ethics statement

All protocols were approved by the ethics committee of Kyoto University, the Ministry of the Environment, and the Forestry Agency of Japan, and adhered to Guidelines for the Use of Animals of the Association for the Study of Animal Behaviour/Animal Behavior Society[37].

### Reporting summary

Further information on research design is available in the Nature Research Reporting Summary linked to this article.

### Data availability

All data used in this study are available in Figshare (https://doi.org/10.6084/m9.figshare.18007046)[38]. Source data are provided with this paper.

### Code availability

R codes used for statistical analysis are available in Figshare (https://doi.org/10.6084/m9.figshare.18007046)[38].

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

## Acknowledgements

We are very grateful to Dr. David Wheatcroft and Dr. Nora V. Carlson for their invaluable comments on the manuscript. This work was supported by JSPS KAKENHI (Grant Numbers JP20H05001 and JP20H03325 to T.N.S. and JP19J01718 to Y.K.M.), the Hakubi Project Funding of Kyoto University (T.N.S.), and JST FOREST Program (Grant Number JPMJFR2149 to T.N.S.).

## Author contributions

T.N.S. conceived the study and drafted the manuscript. T.N.S. and Y.K.M. performed the experiment, analysed the data, and finalized the manuscript.

## Competing interests

The authors declare no competing interests.
