## [Peer Review File · Nature Communications]

Experimental evidence for core-Merge in the vocal communication system of a wild passerineReviewers' Comments:

Reviewer #1:

Remarks to the Author:

This study investigates whether the understanding of call combinations in the Japanese tits is underlain by MERGE, i.e., whether the listeners perceive the alert and recruitment calls as a call combination or as two calls in close temporal proximity. To this end, the authors broadcasted a large set of acoustic stimuli to wild Japanese tits, coming either from one or two speakers, and compared the approach and display behaviors of the birds to a predator model. Since birds approached and displayed more when the stimulus was coming from one speaker compared to two speakers, they concluded that Japanese tits used MERGE to process call combinations.

This review will be rather short, as this is the first time that I have this little to suggest to improve a paper. This study is needed and timely, for there are growing evidence of syntactic-like structures in animal communication and one of the main critics is that we are not sure whether the signals are emitted/perceived as independent signals happening near to each other, or as true combinations. The methods are incredibly well thought and planned: The authors tested whether the number of speakers could influence the behavioural reaction of the birds toward a predator, and they controlled for the order of the combination, the number of call providers, and pseudo replication, which led them to create a big and balanced set of well-designed stimuli. The results are strong and clear, supported by good visual supports like the supplementary tables and the Fig 4. Finally, the authors did an excellent job at writing a clear, easy to read and concise article. They also designed nice figures that can be very helpful to a more naive reader. For all these reasons, I don't have much to add to improve this paper, as, in my opinion, it is already nearly perfect.

I have a few minor comments:

-L217: it could be clearer if you spoke about the rate of "combination of calls", otherwise it may sound like alert and recruitment calls are separated by 3s – even if that is clarified later.

-L265: it is not clear to me what was randomized within the blocks, since, if I understood well, each block is composed of unique stimuli corresponding to a specific combination of treatment and call sources. Did you mean that the trials were conducted in a randomized order?

-L279: How did you decide on the 2-m criterion?

-L 177: You could cite this paper as a "but see", because they draw different conclusions from the putty-nosed monkeys' dataset: Schlenker, P., Chemla, E., Arnold, K., & Zuberbühler, K. (2016). Pyow-hack revisited: Two analyses of Putty-nosed monkey alarm calls. *Lingua*, 171, 1-23.

-L192: You could cite the following paper, which supports your point that the Japanese tits system (and all animal systems studied so far) only combine two units, compared to human language: Miyagawa S and Clarke E (2019) Systems Underlying Human and Old World Monkey Communication: One, Two, or Infinite. *Front. Psychol.* 10:1911. doi: 10.3389/fpsyg.2019.01911

-L296 and 299: The figshare links do not work!

Reviewer #2:

Remarks to the Author:

This manuscript presents a well-thought out and well-executed (apart from one potential caveat, see below) study on whether Japanese tits distinguish between calls from a single source versus from multiple sources. In particular it shows that Japanese tits have a different reaction when call elements come from sources with different spatial locations than when they come from the same spatial source.

This work certainly has significance for the study of bird behavior, and as far as I can judge it is original and acknowledges and links the existing literature well.

As far as the conclusions go, this work establishes the relevance of combined calls coming from a

single source and the ability of Japanese tits to perceive whether calls come from a single source. I have more difficulty following the link to language evolution and Merge.

The problem probably stems from the fact that the authors do not define what they mean by core-Merge. As they appear to refer to "Merge" without core- (on line 187) as a distinct ability, and given the discussion that follows I assume that they do not mean fully recursive Merge, but just the ability to combine two elements (without then using that combined element as a new element on which Merge can operate). Hence: "...core-Merge is different from recursion..." (line 189).

So what they establish is **not** that Japanese tits react differently on the combined calls than on the separate elements (for this they provide reference 10) but that they do so more strongly if the call elements come from a single source. However, whether this is a more interesting "precursor" (not in the sense of evolutionary homology, but analogy) appears to be a matter of taste. A minimalist linguist would probably argue that the ability to do merge (recursive or not) would not depend on the source producing the words/morphemes, so from that perspective showing the call elements have to come from the same source would be weaker support for non-recursive or core-Merge than if they have to come from the same source.

As for the relation to language evolution, it is interesting to demonstrate that non-recursive Merge-like behaviors exist in other species, indicating that there might be a gradual pathway to recursion, and that macro-mutations are unnecessary in the evolution of language. However, for this argument it also appears unnecessary to show that the call elements that are combined need to come from the same source.

I therefore do not agree with the authors conclusion that "Based on these findings we conclude that tits have evolved core-Merge, which enables them to recognize an alert-recruitment call sequence as a single unit." Why not just say "Based on these findings we conclude that tits have evolved the ability to recognize an alert-recruitment call sequence as a single unit."? I would even go as far to say that the paper would be improved if most of the discussion of core-Merge would be removed (except for the part where it is discussed that the fact that some animals have an ability to combine two elements but no recursion might be an indication that recursion could evolve gradually).

Still, this study is an elegant demonstration of how Japanese tits process call combinations, and it adds to a growing body of literature (relevant to language evolution) that demonstrates combinatorial communication in animals.

As for the methodological caveat I mentioned above: the authors are apparently not aware that audio over Bluetooth involves lossy compression that alters the spectral properties of the reproduced signal. See for instance <http://www.sereneaudio.com/blog/how-good-is-bluetooth-audio-at-its-best> for a useful demonstration. I do not think that this influences the results of **this** paper in any way. However, although the lossy compression is designed to be difficult to notice for humans, it may have more noticeable effects for other species. For the sake of reproducibility, it would therefore be useful if the authors could specify which codec and at which quality level the sounds were transmitted.

Reviewer #3:

Remarks to the Author:

This paper tests the capability of core-Merge in Japanese tits. The authors use playback experiments to broadcast alert-recruit call combinations from the same, or different, sources to test whether tits perceive alert-recruit calls as a single unit. Tits responded strongly to calls played in the correct order from one speaker, with decreased mobbing to calls produced from different sources, indicating tits understand that the information contained within those call combinations to be linked, not only in time, but by the sender. The authors' work provides support for core-Merge in non-human animals, disputing some theories on the evolution of language.

The authors provide a simple, elegant, and effective study design to test the ability of core-Merge

within their well-studied system. This creative and novel experimental approach to unpacking core-Merge is not only interesting in itself, provide strong results to support this theory. The figures explain a complex linguistic concept in an easy to understand way. The discussion does an excellent job placing the authors' research within the body of existing knowledge on call combinations as well as language evolution.

Minor concerns

There are some inconsistencies with the use of "animal" vs. "non-human animal". While understandable in the abstract per word limits, we recommend using non-human animals consistently throughout the paper.

Specific comments:

Line 75-79. The first part of the sentence is vague (which factors?) and we recommend that the authors explicitly state the main factors their design controlled for or minimized (e.g. pseudoreplication).

Line 89-91. The authors did not quantify alarm calling by flocks in response to the stimuli, although the example captured on video includes alarm calls. Do the first tits to arrive give additional alert-recruitment calls to bring in the rest of the flock? Or do they give other alarm calls once the predator is detected? We imagine there's good reason that the calls were not quantified and a short explanation is all that would be needed to provide insight.

Lines 112-115. Split up this long sentence. End sentence after Fig 3c. Then "The calling rate and the duration between two calls types were identical to those of one-speaker playback of alert-recruitment sequences, but the calls types were not presented in the naturally ordered sequence."

line 122-125. In the absence of alarm calls, we might still expect mobbing of a stationary predator. It's curious that the shrike stimulus elicited little response except during the playback of alert-recruitment calls from one speaker. Is it because tits did not detect the model when the incorrect sequence or calls from two locations were played? The authors quantified tits within 2 m of the model and it's unclear whether flocks were attracted at all during the other playbacks. If the birds failed to detect the stimulus in the absence of alert-recruitment calls, this seems worth mentioning to strengthen the argument. Lines 143 and 251 state that data were collected on flock size within 15-m of the model during playback, suggesting that data are at hand.

Lines 135-6. We appreciate that the authors considered call variation in their design. Clarify here that there is replication of individual calls; at present, one could interpret calls originating from only 1-2 individuals total, which is not the case.

Line 155-156. Reword: "tits discriminate between two temporally and spatially linked calls from one speaker, which mimic calls by one individual, and two temporally linked calls played from two speakers, which mimic calls from two individuals."

Line 179-183. Break down the difference between core-Merge and recursion. We think the point the authors are trying to stress here would be more effective with an example of recursion, or a more fleshed out description of the difference between the two theories of language productivity.

Line 259. Reword: "for the two behavioral measurements"

Line 260. Does this mean that the observers moved into the experimental arena between playbacks? Were tits present at this time?

Line 265. Was there a minimum time between successive playbacks?

Line 271. Should this be "unique combinations of exemplars"? Earlier stated that recordings came from 8 individuals.

Line 279. "... tits in flocks THAT (i) approached..."

Line 280-1. "...a fixed term, AND flock size... playbacks) AND playback..."

Reviewer #4:

Remarks to the Author:

Thanks for the opportunity to review the manuscript titled: Experimental evidence for core-Merge in the vocal communication system of Japanese tits.

The manuscript presents experimental evidence for the key idea that two distinct (functionally referential) calls, in this case, alert and recruitment, can recognise by an individual when it comes from a single speaker (as a single unit) than individual components separately from two different speakers (as separate units). Authors proposed a new paradigm to link Japanese tit's alert-recruitment call sequences to core-merge, a cognitive capacity found in human language. Behavioural responses of Japanese tits towards playback speakers and the taxidermic models were used to test the prediction.

The approach is novel, and the manuscript is well written. The manuscript organization is also clear; however, the experimental procedure requires clarity, sufficient details, and clarification before agreeing on conclusions.

General comments:

01. The playback stimuli used in previous experiments and authors used the 'D' notes for recruitment calls and the three other notes for alert calls (Suzuki et al. 2016). The stimulus preparation is important as the repeated 'D' notes can produce against different predators (Suzuki, 2004). However, it was not explicitly said that the alert calls used in the stimuli preparation were specifically recorded, exposing the taxidermic bull-headed shrike specimens (hereafter taxidermic model) to Japanese tits' or natural calls directed towards bull-headed shrikes. If not, do the unique playback exemplars contain different 'D' notes? If yes, details may be helpful for the reader and please include them in the manuscript (Page 11, line 217 or elsewhere).

02. One of the major issues in the experiment is presenting a taxidermic model and playback stimuli because the model can attract Japanese tit individuals by itself. Besides, the attracted individuals either to the taxidermic model or playback can produce predator specific recruitment calls (i.e., more D notes or ABC notes or any other combination), which may overlap with the experimental playbacks during the 90s. It is hard to imagine a mobbing event that attracts individuals do not produce recruitment or alert calls. Have the authors encountered such situations in the trials, and if yes, how many?

03. Authors made a specific effort to control the directionality of the sound source; however, it may not be fully controlled as the individuals can approach above the speakers; in that case, their responses may differ from the ones come from the sides in the same height to the speakers. In addition, the individual auditory and visual perception may not be the same due to the initial spatial location(s) of the approaching bird(s) as the number of birds and the initial distances vary in different trials.

There is an apparent confusion about whether the authors used the distance of 2-m from the taxidermic shrike specimens (page 5, line 89 -91) or 2-m from the speaker (page 13, line 252 -253). This led to several problems or confusion in the data analyses and interpretation of the experimental data.

05. Further, why is an arbitrary 2-m distance used to measure the propensity? Would the propensity change when using different thresholds (i.e., 3m, 4m, 5m)? This measurement was again problematic with two-speaker vs one-speaker comparisons. For example, if a recruitment call attracted an individual close to a 2-m speaker in two-speaker setups, it may be well beyond the 2-m distance from the taxidermic model as the distances were measured from the speaker (page 13, line 252 -253) if that is the case why use a taxidermic model in this experiment. The measurements should be from the taxidermic model described in the introduction (page 5, line 89 -91). Including boxplots of approached distances to the taxidermic model under each treatment in the supplementary material would be better.

06. I wonder why the authors did not use the second speaker in the treatment 1A-R and 1R-A (Fig 3 a and c) using stereo tracks, same as the treatment 2A-R and 2R-A (Fig 3 b and d) instead mono tracks. This experimental layout still answers the question as the combined calls (1A-R and 1R-A) come from a single sound source if the inter-call distances are kept constant between two stereo tracks. Moreover, the authors need to reveal the playback device or devices used in these experiments because two Bluetooth speakers were used for stereo playbacks (page 13, line 245).

07. Adding two individuals in Fig 3 (a and c) is misleading as the single speaker playback did not imply two individuals. In all cases, the individual response is mainly dependent on the spatial location of each Japanese tit before starting the experiment trials, which did not adequately control in the experiment (i.e., lure Japanese tits to a feeding station from a specific distance to speakers and taxidermic models)

It would have been helpful to include spatial speaker arrangement as a diagram in the supplementary materials because speakers from the taxidermic model could be placed in a triangular arrangement, keeping 5m distance and separating speakers 10-m distance OR opposite sides relative to the specimen as in Figure 3 b and d.

08. Regardless of how the 2-m distance was set, either from the taxidermic model or the speakers, the GLMM models used the proportion/propensity of individuals as the response variable. The proportion/propensity is calculated from the number of individuals within a 2-m / group size (number of individuals within 15 m). The number of individuals within 15 m again used as a weight in the same model seems problematic because it is already used to calculate the response variable: propensity. ("/" implies division).

The alternative statistical method may include the number of individuals present within the 2-m radius (please see above comments 03 and 05) as the response variable and then use group size as a weight to fit either Poisson or Negative binomial model.

09. It is unclear how the authors measured the wing flicking behaviour as on page 5, line 91 indicated that wing flicking measure based on the individuals within 15 m and page 13 line 253 implied that percentage of approached birds within 2-m to taxidermic model showed wing flicking behaviour. If the latter was the case, both measures might be correlated because it is highly likely that approached birds also showed wing flicking behaviour.

10. It would also be interesting to know why the authors did not present the predicated probabilities with confidence intervals instead of raw data in Fig 4. The data and code couldn't access through the Figshare link provided (<https://doi.org/10.6084/m9.figshare.18007046>) to review thoroughly.

Minor comments

Page 2, line 21-22: It may be helpful to mention what "temporally-linked" means here or elsewhere. In this context, it is imperative as the manuscript describes "two call types" (page 3, line 39) or "two-call sequences" (page 3, line 41).

Page 11, line 219 -220: please include the details (mean \pm SD) of inter-call duration between two 1A-R or two 1 R-As. Is the inter-call duration between two 2A-R or two 2R-A also the same?

Reviewer #5:

None

RESPONSE TO REVIEWER COMMENTS:

Reviewer #1

This study investigates whether the understanding of call combinations in the Japanese tits is underlain by MERGE, i.e., whether the listeners perceive the alert and recruitment calls as a call combination or as two calls in close temporal proximity. To this end, the authors broadcasted a large set of acoustic stimuli to wild Japanese tits, coming either from one or two speakers, and compared the approach and display behaviors of the birds to a predator model. Since birds approached and displayed more when the stimulus was coming from one speaker compared to two speakers, they concluded that Japanese tits used MERGE to process call combinations.

This review will be rather short, as this is the first time that I have this little to suggest to improve a paper. This study is needed and timely, for there are growing evidence of syntactic-like structures in animal communication and one of the main critics is that we are not sure whether the signals are emitted/perceived as independent signals happening near to each other, or as true combinations. The methods are incredibly well thought and planned: The authors tested whether the number of speakers could influence the behavioural reaction of the birds toward a predator, and they controlled for the order of the combination, the number of call providers, and pseudo replication, which led them to create a big and balanced set of well-designed stimuli. The results are strong and clear, supported by good visual supports like the supplementary tables and the Fig 4. Finally, the authors did an excellent job at writing a clear, easy to read and concise article. They also designed nice figures that can be very helpful to a more naive reader. For all these reasons, I don't have much to add to improve this paper, as, in my opinion, it is already nearly perfect.

Thank you very much for the valuable time and effort you have put into our manuscript and for your very positive comments. We have carefully revised the manuscript in accordance with your comments. Below are our point-by-point responses to your comments.

Please note that line numbers in this letter correspond to those of the word file (see Source File (DOCX)).

I have a few minor comments:

-L217: it could be clearer if you spoke about the rate of “combination of calls”, otherwise it may sound like alert and recruitment calls are separated by 3s – even if that is clarified later.

Thank you for this comment. We clarified the number of calls (call rate) in Lines 233-235.

-L265: it is not clear to me what was randomized within the blocks, since, if I understood well, each block is composed of unique stimuli corresponding to a specific combination of treatment and call sources. Did you mean that the trials were conducted in a randomized order?

Thank you for this clarification. We revised the text to clarify this point. The order of trials was randomized within each block that is composed of unique call exemplars but includes four treatments differing in the number of speakers and call order (i.e., 1A-R, 2A-R, 1R-A, 2R-A; Lines 296-298). We also added Supplemental Table 3 to clarify the sound preparation design (see also Lines 143-149, 256-263).

-L279: How did you decide on the 2-m criterion?

We decided the 2-m criterion because the distance between a shrike specimen and each speaker was 5-m. If we use 3-m criterion, it is difficult to know whether individuals approached to shrikes or speakers. We revised Fig. 3 to make readers easy to understand this experimental set-up.

-L 177: You could cite this paper as a “but see”, because they draw different conclusions from the putty-nosed monkeys’ dataset: Schlenker, P., Chemla, E., Arnold, K., & Zuberbühler, K. (2016). Pyow-hack revisited: Two analyses of Putty-nosed monkey alarm calls. *Lingua*, 171, 1-23.

Agreed. We added this paper accordingly (ref. 21, Line 189).

-L192: You could cite the following paper, which supports your point that the Japanese tits system (and all animal systems studied so far) only combine two units, compared to human language: Miyagawa S and Clarke E (2019) Systems Underlying Human and Old World Monkey Communication: One, Two, or Infinite. *Front. Psychol.* 10:1911. doi: 10.3389/fpsyg.2019.01911

Thank you. We added this citation, as suggested (ref.29, Line 210).

-L296 and 299: The figshare links do not work!

Figshare links will be available when the paper is published online.

Reviewer #2

This manuscript presents a well-thought out and well-executed (apart from one potential caveat, see below) study on whether Japanese tits distinguish between calls from a single source versus from multiple sources. In particular it shows that Japanese tits have a different reaction when call elements come from sources with different spatial locations than when they come from the same spatial source.

This work certainly has significance for the study of bird behavior, and as far as I can judge it is original and acknowledges and links the existing literature well.

Thank you very much for your time, effort, and positive comments on the earlier version of this manuscript. We have now carefully revised the manuscript according to the suggestions. Please find below our point-by-point responses to your comments.

Please note that line numbers in this letter correspond to those of the word file (see Source File (DOCX)).

As far as the conclusions go, this work establishes the relevance of combined calls coming from a single source and the ability of Japanese tits to perceive whether calls come from a single source. I have more difficulty following the link to language evolution and Merge.

The problem probably stems from the fact that the authors do not define what they mean by core-Merge. As they appear to refer to "Merge" without core- (on line 187) as a distinct ability, and given the discussion that follows I assume that they do not mean fully recursive Merge, but just the ability to combine two elements (without then using that combined element as a new element on which Merge can operate). Hence: "...core-Merge is different from recursion..." (line 189).

So what they establish is *not* that Japanese tits react differently on the combined calls than on the separate elements (for this they provide reference 10) but that they do so more strongly if the call elements come from a single source. However, whether this is a more interesting "precursor" (not in the sense of evolutionary homology, but analogy) appears to be a matter of taste. A minimalist linguist would probably argue that the ability to do merge (recursive or not) would not depend on the source producing the words/morphemes, so from that perspective showing the call elements have to come from the same source would be weaker support for non-recursive or core-Merge than if they have to come from the same source.

As for the relation to language evolution, it is interesting to demonstrate that non-recursive Merge-like behaviors exist in other species, indicating that there might be a gradual

pathway to recursion, and that macro-mutations are unnecessary in the evolution of language. However, for this argument it also appears unnecessary to show that the call elements that are combined need to come from the same source.

I therefore do not agree with the authors conclusion that "Based on these findings we conclude that tits have evolved core-Merge, which enables them to recognize an alert-recruitment call sequence as a single unit." Why not just say "Based on these findings we conclude that tits have evolved the ability to recognize an alert-recruitment call sequence as a single unit."? I would even go as far to say that the paper would be improved if most of the discussion of core-Merge would be removed (except for the part where it is discussed that the fact that some animals have an ability to combine two elements but no recursion might be an indication that recursion could evolve gradually).

Still, this study is an elegant demonstration of how Japanese tits process call combinations, and it adds to a growing body of literature (relevant to language evolution) that demonstrates combinatorial communication in animals.

We agreed that the definitions of "Merge" or "core-Merge" should be clarified. We explained the definition of Merge and core-Merge in Introduction (Lines 31-36) and Discussion (Lines 198-209). We defined "Merge" as a cognitive capacity to combine two linguistic items (e.g., two words or two phrases) into a sequence and to recognize it as a new unit, while "core-Merge" refers to the cognitive capacity to produce and perceive the most basic combinations of meaningful items (words) as a single unit. Our previous studies showed that Japanese tits combine two meaningful calls into a sequence and the current study showed that receivers recognize the sequence as a single unit. Therefore, we keep our conclusion that our data provide evidence for core-Merge in the Japanese tit.

We agree that minimalist linguists might claim that humans may recognize the two words as a single phrase even if they come from two sources. For example, if A says "good" and B says "morning" in close time proximity, listeners may understand the two words as a single phrase "good morning", due to the capacity of core-Merge. However, when studying animals, it is often necessary to find a paradigm that allows us to more robustly test if a given response is produced in response to a stimulus under specific conditions (e.g., signals produced by one individual combining two calls into one unit) or under arbitrary conditions (e.g., two calls produced arbitrarily that happen to be temporally-linked). This lets us create circumstances closer to those found in human language such as the one explained in Fig. 1, where listeners recognize two words as a single, merged unit only when they are produced by a single person. For example, if a single person says "come talk", this is interpreted as a message of come and talk, while the same two words are

separately given by two persons (A says “come”, B says “talk”), these words could be interpreted as two individual messages. This experiment creates a context where core-Merge would only be occurring if the signaler is combining two call types to form a third, unique, call unit, and not when two individuals are producing single calls on their own. While in human linguistic two words produced by two separate humans may be able to be perceived as one work when they work together, there is no evidence that Japanese tits do this (though some chorusing species do work together vocally like this). This paradigm allows us to test if these birds are combining two separate calls into a unit (i.e. if this species has the capacity for core-Merge) by using a specific context. The differences in response in these two contexts provide evidence for core-Merge (information transmission by temporal and spatial link of two items, which functions as a single unit).

Thus, for non-human animals, if we find at least one case where receivers respond differently to two temporally-linked calls produced by a single individual and to two temporally-linked calls produced by two individuals, we can reject the possibility that receivers recognize two words as two individual messages that are simply linked in time, but do interpret them as a larger unit, providing evidence for core-Merge.

As for the methodological caveat I mentioned above: the authors are apparently not aware that audio over Bluetooth involves lossy compression that alters the spectral properties of the reproduced signal. See for instance <http://www.sereneaudio.com/blog/how-good-is-bluetooth-audio-at-its-best> for a useful demonstration. I do not think that this influences the results of *this* paper in any way. However, although the lossy compression is designed to be difficult to notice for humans, it may have more noticeable effects for other species. For the sake of reproducibility, it would therefore be useful if the authors could specify which codec and at which quality level the sounds were transmitted.

We used Bluetooth speakers so as to fix the experimental set-up easily and quickly. This was so important as we needed to set up speakers and a predator specimen as soon as possible once we detected Japanese tits. Otherwise, Japanese tits leave the area before starting experiments. As you thought, usage of Bluetooth speakers does not influence the results or conclusions of this paper, but we will keep this in our mind for the future study. Thank you for this information.

Reviewer #3

This paper tests the capability of core-Merge in Japanese tits. The authors use playback experiments to broadcast alert-recruit call combinations from the same, or different, sources to test whether tits perceive alert-recruit calls as a single unit. Tits responded strongly to calls played in the correct order from one speaker, with decreased mobbing to calls produced from different sources, indicating tits understand that the information contained within those call combinations to be linked, not only in time, but by the sender. The authors' work provides support for core-Merge in non-human animals, disputing some theories on the evolution of language.

The authors provide a simple, elegant, and effective study design to test the ability of core-Merge within their well-studied system. This creative and novel experimental approach to unpacking core-Merge is not only interesting in itself, provide strong results to support this theory. The figures explain a complex linguistic concept in an easy to understand way. The discussion does an excellent job placing the authors' research within the body of existing knowledge on call combinations as well as language evolution.

Thank you very much for your time, effort, and very positive comments on the earlier version of our manuscript. We have now carefully revised the manuscript according to the suggestions. Please find below our point-by-point responses to your comments.

Please note that line numbers in this letter correspond to those of the word file (see Source File (DOCX)).

Minor concerns

There are some inconsistencies with the use of "animal" vs. "non-human animal". While understandable in the abstract per word limits, we recommend using non-human animals consistently throughout the paper.

Agreed. We replaced "animals" with "non-human animals" as possible. However, it seems to be redundant if we add "nonhuman" in all the places. So, we explained "animals" refer to "nonhuman animals" in Line 37.

Specific comments:

Line 75-79. The first part of the sentence is vague (which factors?) and we recommend that the authors explicitly state the main factors their design controlled for or minimized (e.g. pseudoreplication).

Agreed. We revised this sentence to clarify how we made efforts when making playback files (Lines 80-86, see also Lines 296-298).

Line 89-91. The authors did not quantify alarm calling by flocks in response to the stimuli, although the example captured on video includes alarm calls. Do the first tits to arrive give additional alert-recruitment calls to bring in the rest of the flock? Or do they give other alarm calls once the predator is detected? We imagine there's good reason that the calls were not quantified and a short explanation is all that would be needed to provide insight.

We observed some cases in which birds produced vocalizations during experimental trials, but it was difficult to count the exact number of individuals that produced vocalizations during the trials. Therefore, we did not use the proportion of individuals that produced calls as a behavioral variable. While some birds produced calls in the close distance from the predator, others produced calls far from the predator specimen.

In consideration for the influence of a particular individual (the first bird) on other flock members, we have added to this revision an analysis on whether at least one bird approached the shrike and exhibited wing flicks during 90-s of playbacks. This analysis can minimize the influence of the first tits' behavior, including counter-calling, since the data come from the first responders. We provided this analysis as a supplemental information (see Supplementary Fig. 1 and Supplementary Table 2), as we think the proportion of flock members (Fig. 4) is still more descriptive and informative, and with the raw data points (blue circles), it is easy to understand there are many zero values in treatments other than 1A-R.

Lines 112-115. Split up this long sentence. End sentence after Fig 3c. Then "The calling rate and the duration between two calls types were identical to those of one-speaker playback of alert-recruitment sequences, but the calls types were not presented in the naturally ordered sequence."

Corrected accordingly (Lines 123-125).

line 122-125. In the absence of alarm calls, we might still expect mobbing of a stationary predator. It's curious that the shrike stimulus elicited little response except during the playback of alert-recruitment calls from one speaker. Is it because tits did not detect the model when the incorrect sequence or calls from two locations were played? The authors quantified tits within 2 m of the model and it's unclear whether flocks were attracted at all during the other playbacks. If the birds failed to detect the stimulus in the absence of alert-recruitment calls, this seems worth mentioning to strengthen the argument. Lines 143 and 251 state that data were collected on flock size within 15-m of the model during playback, suggesting that data are at hand.

Thank you for this comment. We were thinking of the same analysis before, but we thought it could be ambiguous and problematic. In 1R-A, 2A-R, and 2R-A treatments, there are some cases where birds approached to the shrike specimen within 5-m or even within 2-

m (Fig. 4), while they did not exhibit wing flicking behaviors and just continued foraging. Although we felt they should visually perceive/detect the shrike specimen at such a close distance, without any specific behavior that indicates the visual detection of shrikes, it was really difficult to determine whether these birds had visually detected the predator specimen.

Without adding this possibly ambiguous analysis, we can conclude from our results that tits mob a predator when and only when hearing alert-recruitment call sequences produced by a single speaker. The detailed cognitive processes how birds start predator mobbing is the matter for future investigations.

Lines 135-6. We appreciate that the authors considered call variation in their design. Clarify here that there is replication of individual calls; at present, one could interpret calls originating from only 1-2 individuals total, which is not the case.

Thank you for this comment. We clarified this by adding the number of individuals that were used to construct the playback stimuli (Lines 143-149, 256-263) as well as a new supplemental table (Supplementary Table 3).

Line 155-156. Reword: "tits discriminate between two temporally and spatially linked calls from one speaker, which mimic calls by one individual, and two temporally linked calls played from two speakers, which mimic calls from two individuals."

Corrected based on this comment (Lines 165-167).

Line 179-183. Break down the difference between core-Merge and recursion. We think the point the authors are trying to stress here would be more effective with an example of recursion, or a more fleshed out description of the difference between the two theories of language productivity.

Thank you for this helpful comment. We revised this paragraph according to this comment and the comments by Reviewer 2 (Lines 198-209).

Line 259. Reword: "for the two behavioral measurements"

Corrected (Line 289).

Line 260. Does this mean that the observers moved into the experimental arena between playbacks? Were tits present at this time?

After the completion of each trial at each experimental site, observers checked the video recording and made an on-the-spot confirmation of the exact location at which each bird made the closest approach to the shrike specimen during 90-s of playbacks. Tits typically left the playback site during or soon after the playbacks, and thus they were not present. We clarified this in Lines 290-292.

Line 265. Was there a minimum time between successive playbacks?

We did not make any minimum time between successive playbacks. Instead, we ensured the data independency by separating closest experimental sites by at least 400-m apart (Lines 218-221).

Line 271. Should this be “unique combinations of exemplars”? Earlier stated that recordings came from 8 individuals.

We revised this sentence (Lines 304-305) and clarified how we prepared playback stimuli (Lines 143-149, 256-263, and Supplementary Table 3).

Line 279. “... tits in flocks THAT (i) approached...”

Corrected (Line 312).

Line 280-1. “...a fixed term, AND flock size... playbacks) AND playback...”

Corrected (Lines 313-314).

Reviewer #4

Thanks for the opportunity to review the manuscript titled: Experimental evidence for core-Merge in the vocal communication system of Japanese tits.

The manuscript presents experimental evidence for the key idea that two distinct (functionally referential) calls, in this case, alert and recruitment, can be recognised by an individual when it comes from a single speaker (as a single unit) than individual components separately from two different speakers (as separate units). Authors proposed a new paradigm to link Japanese tit's alert-recruitment call sequences to core-merge, a cognitive capacity found in human language. Behavioural responses of Japanese tits towards playback speakers and the taxidermic models were used to test the prediction.

The approach is novel, and the manuscript is well written. The manuscript organization is also clear; however, the experimental procedure requires clarity, sufficient details, and clarification before agreeing on conclusions.

Thank you very much for your time, effort, and positive comments on the earlier version of this manuscript. We have now carefully revised the manuscript according to the suggestions. Please find below our point-by-point responses to your comments.

Please note that line numbers in this letter correspond to those of the word file (see Source File (DOCX)).

General comments:

01. The playback stimuli used in previous experiments and authors used the 'D' notes for recruitment calls and the three other notes for alert calls (Suzuki et al. 2016). The stimulus preparation is important as the repeated 'D' notes can produce against different predators (Suzuki, 2004). However, it was not explicitly said that the alert calls used in the stimuli preparation were specifically recorded, exposing the taxidermic bull-headed shrike specimens (hereafter taxidermic model) to Japanese tits' or natural calls directed towards bull-headed shrikes. If not, do the unique playback exemplars contain different 'D' notes? If yes, details may be helpful for the reader and please include them in the manuscript (Page 11, line 217 or elsewhere).

Thank you for this clarification. We conducted predator exposure experiments to Japanese tit flocks and recorded mobbing calls towards a specimen of the bull-headed shrike. In response to a shrike specimen, tits produced alert-recruitment call sequences and the note repetition number of recruitment call part (D notes) ranged from 5 to 15. Since the interquartile range of repetition number was 6.75 to 10, we used recruitment calls with 7-10 notes as playback stimuli in this study. We added this information in Lines 241-248.

02. One of the major issues in the experiment is presenting a taxidermic model and playback stimuli because the model can attract Japanese tit individuals by itself. Besides, the attracted individuals either to the taxidermic model or playback can produce predator specific recruitment calls (i.e., more D notes or ABC notes or any other combination), which may overlap with the experimental playbacks during the 90s. It is hard to imagine a mobbing event that attracts individuals do not produce recruitment or alert calls. Have the authors encountered such situations in the trials, and if yes, how many?

Exposure of a predator specimen in combination with playback stimuli allowed us to measure tits' mobbing responses during one- and two-speaker playbacks through a common standard (Lines 92-94). If we do not use the taxidermic model, we cannot compare tits approaching response (one of the most important components of birds' mobbing behaviour) between one- and two-speaker playbacks (i.e., the taxidermic mount serves as the target of approaching (or mobbing), even if the number of speakers is different). We did not observe cases that exposing a shrike solely attracted birds to the arena. All the birds approached to the area only after playbacks commenced, and before the playbacks they were foraging around 15-m away from the shrike specimen.

We observed some cases in which birds produced vocalizations during experimental trials, but it was difficult to count the exact number of individuals within a flock that produced vocalizations during experiments. Therefore, we did not use the proportion of individuals that produced calls as a behavioral variable. While some birds produced calls in the close distance from the predator, others produced calls far from the predator specimen.

In consideration for the influence of a particular individual (the first bird) on other flock members, we have added to this revision an analysis on whether at least one bird approached the shrike and exhibited wing flicks during 90-s of playbacks. This analysis can minimize the influence of the first tits' behavior, including counter-calling, since the data come from the first responders. We provided this analysis as a supplemental information (see Supplementary Fig. 1 and Supplementary Table 2), as we think the proportion of flock members (Fig. 4) is still more descriptive and informative, and with the raw data points (blue circles), it is easy to understand there are many zero values in treatments other than 1A-R.

03. Authors made a specific effort to control the directionality of the sound source; however, it may not be fully controlled as the individuals can approach above the speakers; in that case, their responses may differ from the ones come from the sides in the same height to the speakers. In addition, the individual auditory and visual perception

may not be the same due to the initial spatial location(s) of the approaching bird(s) as the number of birds and the initial distances vary in different trials.

In the field setting, it is impossible to fully control the spatial location of free-living individuals. So, we made our best effort to standardize this: we directed speakers upward and started playbacks only when the flock was ca. 15-m from the shrike specimen (Lines 274-276, 281-282).

04. There is an apparent confusion about whether the authors used the distance of 2-m from the taxidermic shrike specimens (page 5, line 89 -91) or 2-m from the speaker (page 13, line 252 -253). This led to several problems or confusion in the data analyses and interpretation of the experimental data.

Thank you for finding this typo. We measured the response of tits towards “a shrike specimen”, but not towards “speakers”. So, “2-m from the speaker” should be “2-m of the shrike specimen”. We revised this phrase accordingly (Line 283).

05. Further, why is an arbitrary 2-m distance used to measure the propensity? Would the propensity change when using different thresholds (i.e., 3m, 4m, 5m)? This measurement was again problematic with two-speaker vs one-speaker comparisons. For example, if a recruitment call attracted an individual close to a 2-m speaker in two-speaker setups, it may be well beyond the 2-m distance from the taxidermic model as the distances were measured from the speaker (page 13, line 252 -253) if that is the case why use a taxidermic model in this experiment. The measurements should be from the taxidermic model described in the introduction (page 5, line 89-91). Including boxplots of approached distances to the taxidermic model under each treatment in the supplementary material would be better.

We realized this comment was derived from our typo that was corrected in the previous comment 04 and apologized leading this reviewer to confusion. We measured approaching distance from a shrike specimen (but not from speakers), as supposed later in this comment. We used a shrike specimen since it enables us to compare the response of birds between treatments that even differ in the number of speakers. We chose 2-m distance, because if we use 3-m distance, it was not clear whether birds approached to the shrike specimen or speakers (as the shrike and each speaker was separated by 5-m, approaching within 2-m from the shrike would be appropriate for measure approaching behavior). To make this point easy to understand, we added the distance between speakers and a shrike specimen in Fig. 3.

Thank you for considering a figure with boxplots of approach distances. However, it is difficult to measure approach distances to the shrike specimen for all treatments; in treatments other than 1A-R, most birds did not approach to 2-m from the shrike and thus, it was hard to exactly measure the distance between these birds and the shrike specimen. See details of how we measured the exact approach distance in Lines 290-295.

06. I wonder why the authors did not use the second speaker in the treatment 1A-R and 1R-A (Fig 3 a and c) using stereo tracks, same as the treatment 2A-R and 2R-A (Fig 3 b and d) instead mono tracks. This experimental layout still answers the question as the combined calls (1A-R and 1R-A) come from a single sound source if the inter-call distances are kept constant between two stereo tracks. Moreover, the authors need to reveal the playback device or devices used in these experiments because two Bluetooth speakers were used for stereo playbacks (page 13, line 245).

This is because if we use two speakers for 1A-R and 1R-A treatments, the calling rate will be half for each speaker in compared with 2A-R and 2R-A. So, by using a single speaker for 1A-R and 1R-A treatments, we could ensure that only the spatial link between A (alert call) and R (recruitment call) is different between one and two speaker treatments while calling rate is constant across treatments (Lines 84-86).

07. Adding two individuals in Fig 3 (a and c) is misleading as the single speaker playback did not imply two individuals. In all cases, the individual response is mainly dependent on the spatial location of each Japanese tit before starting the experiment trials, which did not adequately control in the experiment (i.e., lure Japanese tits to a feeding station from a specific distance to speakers and taxidermic models).

It would have been helpful to include spatial speaker arrangement as a diagram in the supplementary materials because speakers from the taxidermic model could be placed in a triangular arrangement, keeping 5m distance and separating speakers 10-m distance OR opposite sides relative to the specimen as in Figure 3 b and d.

Thank you so much for this comment. We redrew Fig. 3 not to make readers misleading. It is impossible to control for the spatial location of each bird in a field setting; even if we use feeding stations, birds within a flock would feed at diverse spatial locations, such as on trees or on the ground. In addition, if we use feeding stations, inter-individual aggressions for clumped food were unnaturally induced. So, we tried to control for the location by beginning the experiments only when at least two individuals were ca. 15-m from the shrike specimen (Lines 281-282).

We set distance between each speaker and shrike specimen by 5-m and also set distance between two speakers by 10-m. So, it is not possible to arrange two speakers and a shrike specimen in a triangular arrangement (it should be always linear arrangement). We thought this is an important point and thus added this information in the main figure (Fig. 3 legend), instead of adding a new supplemental figure as suggested.

08. Regardless of how the 2-m distance was set, either from the taxidermic model or the speakers, the GLMM models used the proportion/propensity of individuals as the response variable. The proportion/propensity is calculated from the number of individuals within a 2-m / group size (number of individuals within 15 m). The number of individuals within 15 m again used as a weight in the same model seems problematic because it is already used to calculate the response variable: propensity. ("/" implies division).

The alternative statistical method may include the number of individuals present within the 2-m radius (please see above comments 03 and 05) as the response variable and then use group size as a weight to fit either Poisson or Negative binomial model.

When analyzing proportion data, R binds together two vectors into a single object, y , comprising the number of successes (number of responding individuals) and the number of failures (number of non-responding individuals), and carries out weighted regression (see Clawley 2007 The R Book). This weighted regression considers for the difference in denominator (i.e., number of individuals within a flock), e.g., $1/2$ and $10/20$ are both 50% (or 0.5), but the latter value ($10/20$) is more important as it has a larger denominator. This regression can be carried out by either inserting “weights” argument in the formula or by using “cbind” in dependent variable. We will share the R code together with the data sheet in Figshare.

09. It is unclear how the authors measured the wing flicking behaviour as on page 5, line 91 indicated that wing flicking measure based on the individuals within 15 m and page 13 line 253 implied that percentage of approached birds within 2-m to taxidermic model showed wing flicking behaviour. If the latter was the case, both measures might be correlated because it is highly likely that approached birds also showed wing flicking behaviour.

We observed birds that approached within 2-m of the shrike specimen but did not exhibit wing flicking displays. So, we considered these two behaviours independently. We measured the number of total individuals (within 15m) that exhibited wing flicking, and the number of individuals that came within 2m out of the total number of individuals in 15 m. We clarified this by replacing the word “they” to “birds” in Line 283.

10. It would also be interesting to know why the authors did not present the predicated probabilities with confidence intervals instead of raw data in Fig 4. The data and code couldn't access through the Figshare link provided (<https://doi.org/10.6084/m9.figshare.18007046>) to review thoroughly.

We prefer to make figs with raw data, as we believe this is more informative for this experimental study (the data included many zero values). Thank you for checking the link of Figshare, which will be available when this paper is published online.

Minor comments

Page 2, line 21-22: It may be helpful to mention what "temporally-linked" means here or elsewhere. In this context, it is imperative as the manuscript describes "two call types" (page 3, line 39) or "two-call sequences" (page 3, line 41).

Corrected wording in abstract (Lines 21ff.) and the main text (Lines 42, 48-49).

Page 11, line 219 -220: please include the details (mean +-SD) of inter-call duration between two 1A-R or two 1 R-As. Is the inter-call duration between two 2A-R or two 2R-A also the same?

We have added the information of between-call sequence intervals (range and median); these intervals were constant for playback stimuli (1A-R, 2A-R, 1R-A, 2R-A) that are composed of the same call exemplars (Lines 238-241).

Reviewers' Comments:

Reviewer #2:

Remarks to the Author:

The authors did a good job of revising their manuscript. With their clarifications, my objections have been addressed.

Reviewer #4:

Remarks to the Author:

Overall, the authors put a great effort into responding to the queries and questions we raised and added more details to the second draft of the manuscript. I thank them for their effort and time. I also thank the authors for providing codes and the data for the second round. If data files and codes were available in the first round, the below comments would have been provided earlier to save time and effort.

Thanks for including the details of the playback equipment type and speaker type for reproducibility. Please include the details if you have used any specific app or software when connected to i phone 8 with two blue tooth speakers (page 14, lines 267 & 275, where appropriate) to playback the specific track (left and right). I had the wrong impression that the video recordings were analysed at the lab after the trials. However, the authors managed to measure distances "on the spot" using the video playbacks, which is an impressive effort (archiving experimental videos are always helpful and is a good practice!).

Statistical analyses presented in the second revision still require verification before acceptance.

Based on the response to comment # 02, the authors added an analysis to the supplementary material, claiming that "This analysis can minimise the influence of the first tits' behavior, including counter-calling, since the data come from the first responders". However, I am not convinced as the analyses were performed by converting the number of birds that approached the taxidermic model as binary data ("merge_binary.R"). Furthermore, if I understood the text and author's responses correctly, no video recordings were available to identify the first approached individual for detailed analysis correctly. Therefore, one can argue that the supplementary analysis simply converted the proportional data to a binary response variable (one or more Japanese tits' presence or absence within 2m)!

The following points may also be helpful for authors further to strengthen both "2m" and "wing" statistical analyses in "merge.proportion.R" and the interpretation of the results in the manuscript.

Based on the response to comment # 08, I agree with the authors that the binomial model can fit in two different ways in R. Thanks for the explanation provided and how R works on the data set. Where success (number of individuals approached 2m) and failure (number of individuals not approached 2m), the number of trials is equivalent to flock size. We should be careful that the R formula/syntax and functions do the same as we expect here. In this study, flock size was not the same at each trial, even within a block. Authors have attempted to control it statistically. However, using the formula/syntax with the "weights" argument does not mean controlling the flock size statistically or correctly applying the weights w.r.t large flock size.

Please see page 3 equation 1 and 2 in Bates, D., Mächler, M., Bolker, B., & Walker, S. (2015). Fitting Linear Mixed-Effects Models Using lme4. *Journal of Statistical Software*, 67(1), 1–48.
<https://doi.org/10.18637/jss.v067.i01>

The glmer weight argument contains known prior weights and goes to the diagonal matrix (Bates et al., 2015). As far as I understood, including flock size as a predictor variable is a problem as the flock size is equivalent to the number of trials in the binomial equation.

For example, the below Syntax1 also produces the same results as Syntax2 without weights argument, authors correctly mentioned in the response letter. Yet, selecting the Syntax2 does not imply that flock size is being controlled or correctly applied prior weights. (See Package lme4, April 7, 2022, page 13).

```
Syntax1: model_2m<- glmer (cbind (X2m, flock_size-X2m) ~ trt + flock_size + call_source + (1|caller_ID) + (1|shrike), family="binomial", data=d)
```

*** Please see the above comment on using flock_size as a predictor variable in this context I am not entirely convinced of the usage below formula (Syntax2) instead of the above (Syntax1) as a "weighted regression "unless the authors have a good reason to believe the below equation is better suited to control for the flock size. Please check the introduction to Weighted Mixed-Effects Models with WeMix package by Paul Bailey, Claire Kelley, and Trang Nguyen in 2019 to add weights to observations.

```
Syntax2: model_2m<- glmer (X2m/flock_size~ trt + flock_size + call_source + (1|caller_ID) + (1|shrike), weights=flock_size, family=binomial("logit"), data=d)
```

*** please check" "is needed for logit function specification in "merge.proportion.R", which may differ based on the lme4 version you have used.

Further, please check the possible model overfitting (singularity issue/ warnings) and overdispersion in all models; for example, overdispersion is high for Syntax1 and Syntax2. If overdispersion is significantly high, I would consider alternative analyses approach or drop flock size as a predictor variable. Finally, if the authors used a model selection method, it would be great to include it in the statistical methods section.

I hope these comments will help to improve the manuscript.

RESPONSE TO REVIEWER COMMENTS:

Reviewer #2

The authors did a good job of revising their manuscript. With their clarifications, my objections have been addressed.

Thank you so much for reviewing our revised manuscript. We are happy to hear that our revisions have satisfied your concerns.

Reviewer #4 (Remarks to the Author):

Overall, the authors put a great effort into responding to the queries and questions we raised and added more details to the second draft of the manuscript. I thank them for their effort and time. I also thank the authors for providing codes and the data for the second round. If data files and codes were available in the first round, the below comments would have been provided earlier to save time and effort.

Thank you so much for reviewing our revised manuscript and for your very positive comments. Below are our point-by-point responses to the remaining issues.

Thanks for including the details of the playback equipment type and speaker type for reproducibility. Please include the details if you have used any specific app or software when connected to i phone 8 with two blue tooth speakers (page 14, lines 267 & 275, where appropriate) to playback the specific track (left and right). I had the wrong impression that the video recordings were analysed at the lab after the trials. However, the authors managed to measure distances "on the spot" using the video playbacks, which is an impressive effort (archiving experimental videos are always helpful and is a good practice!).

Thank you for this comment. We didn't use any specific app to playback bird calls. We directly transferred sound files to our device (iphone) as a file folder and connected to the speaker using Bluetooth.

Statistical analyses presented in the second revision still require verification before acceptance.

Based on the response to comment # 02, the authors added an analysis to the supplementary material, claiming that "This analysis can minimise the influence of the first tits' behavior, including counter-calling, since the data come from the first responders". However, I am not convinced as the analyses were performed by converting the number of birds that approached the taxidermic model as binary data ("merge_binary.R"). Furthermore, if I understood the text and author's responses correctly, no video recordings were available to identify the first approached individual for detailed analysis correctly. Therefore, one can argue that the supplementary analysis simply converted the proportional data to a binary response variable (one or more Japanese tits' presence or absence within 2m)!

Thank you for this comment. As we explained in the previous letter, it was difficult to count

the exact number of calling individuals during experiments. So, we used only (i) predator approach and (ii) wing flicking behavior to describe mobbing responses. It doesn't matter if response of a particular bird (including predator approach, wing flicking, or counter-calling) influences other birds' behavior. What this proportion data analyses show is that only one-speaker playback of alert-recruitment call sequences triggers mobbing behavior at the *group level*, providing evidence for core-Merge.

Although we think proportional analyses are enough for testing our hypothesis, we still think that having binary analyses in supplement is informative. In the binary data analysis, we determined whether at least one bird exhibited each behavioral response (yes: there was a bird exhibiting a response (approach/wing flick) without following other birds' response, no: there was no bird exhibiting a response). Therefore, this can minimize the influence of a particular bird's behavior on subsequent individuals, allowing statistical analyses at the *individual level*.

The following points may also be helpful for authors further to strengthen both "2m" and "wing" statistical analyses in "merge.proportion.R" and the interpretation of the results in the manuscript.

Based on the response to comment # 08, I agree with the authors that the binomial model can fit in two different ways in R. Thanks for the explanation provided and how R works on the data set. Where success (number of individuals approached 2m) and failure (number of individuals not approached 2m), the number of trials is equivalent to flock size. We should be careful that the R formula/syntax and functions do the same as we expect here. In this study, flock size was not the same at each trial, even within a block. Authors have attempted to control it statistically. However, using the formula/syntax with the "weights" argument does not mean controlling the flock size statistically or correctly applying the weights w.r.t large flock size.

Please see page 3 equation 1 and 2 in Bates, D., Mächler, M., Bolker, B., & Walker, S. (2015). Fitting Linear Mixed-Effects Models Using lme4. *Journal of Statistical Software*, 67(1), 1–48. <https://doi.org/10.18637/jss.v067.i01>

The glmer weight argument contains known prior weights and goes to the diagonal matrix (Bates et al., 2015). As far as I understood, including flock size as a predictor variable is a problem as the flock size is equivalent to the number of trials in the binomial equation.

For example, the below Syntax1 also produces the same results as Syntax2 without weights argument, authors correctly mentioned in the response letter. Yet, selecting the Syntax2 does not imply that flock size is being controlled or correctly applied prior weights. (See Package lme4, April 7, 2022, page 13).

```
Syntax1: model_2m<- glmer (cbind (X2m, flock_size-X2m) ~ trt + flock_size + call_source + (1|caller_ID) + (1|shrike), family="binomial", data=d)
```

*** Please see the above comment on using flock_size as a predictor variable in this context

I am not entirely convinced of the usage below formula (Syntax2) instead of the above (Syntax1) as a "weighted regression "unless the authors have a good reason to believe the below equation is better suited to control for the flock size. Please check the introduction to Weighted Mixed-Effects Models with WeMix package by Paul Bailey, Claire Kelley, and Trang Nguyen in 2019 to add weights to observations.

```
Syntax2: model_2m<- glmer (X2m/flock_size~ trt + flock_size + call_source + (1|caller_ID) + (1|shrike), weights=flock_size, family=binomial("logit"), data=d)
```

Thank you for this concern. We have changed Syntax 2 to Syntax 1, as suggested (see codes and Lines 313-318). Please note that there are **NO changes** in the statistical results (indicating that both syntax runs the same analyses). For reference of cbind function for proportion data analyses, we added a citation in the reference list (Crawley 2012 *The R Book*). You may also find examples using cbind function for proportion data in the R documentation for "glmer {lme4}".

*** please check " "is needed for logit function specification in "merge.proportion.R", which may differ based on the lme4 version you have used.

Further, please check the possible model overfitting (singularity issue/ warnings) and overdispersion in all models; for example, overdispersion is high for Syntax1 and Syntax2. If overdispersion is significantly high, I would consider alternative analyses approach or drop flock size as a predictor variable.

We inserted "" in the R code to be appropriate for the latest version of lme4 and updated the R version in the reference list. We didn't find either problematic errors or overdispersion in our analyses (see below, values were calculated using *overdisp.glmer* in *RVAideMemoire*).

Proportional analyses

Predator approach: Residual deviance: 45.75 on 56 degrees of freedom (ratio: 0.817)

Wing flicks: Residual deviance: 48.695 on 56 degrees of freedom (ratio: 0.87)

Binary analyses

Predator approach: Residual deviance: 36.816 on 56 degrees of freedom (ratio: 0.657)

Wing flicks: Residual deviance: 39.852 on 56 degrees of freedom (ratio: 0.712)

Finally, if the authors used a model selection method, it would be great to include it in the statistical methods section.

I hope these comments will help to improve the manuscript.

In this study, we designed experiments to test a clear hypothesis that tits perceive a two-call sequence as a single unit (core-Merge). Therefore, we chose “hypothesis testing approach” instead of “model selection approach”. We don’t prefer to mix these two approaches in the same analysis, as they follow different principles of analyses (p-value based testing vs AIC based parameter searching).

Thank you for your time again. We hope this revision satisfies your comments.

Reviewers' Comments:

Reviewer #4:

None